# Understanding the Role of Duration of Vaccine Protection with MenAfriVac: Simulating Alternative Vaccination Strategies

**DOI:** 10.3390/microorganisms9020461

**Published:** 2021-02-23

**Authors:** Andromachi Karachaliou Prasinou, Andrew J. K. Conlan, Caroline L. Trotter

**Affiliations:** Disease Dynamics Unit, Department of Veterinary Medicine, University of Cambridge, Cambridge CB3 0ES, UK; ajkc2@cam.ac.uk (A.J.K.C.); clt56@cam.ac.uk (C.L.T.)

**Keywords:** meningitis, vaccine, Africa, mathematical modelling

## Abstract

We previously developed a transmission dynamic model of *Neisseria meningitidis* serogroup A (NmA) with the aim of forecasting the relative benefits of different immunisation strategies with MenAfriVac. Our findings suggested that the most effective strategy in maintaining disease control was the introduction of MenAfriVac into the Expanded Programme on Immunisation (EPI). This strategy is currently being followed by the countries of the meningitis belt. Since then, the persistence of vaccine-induced antibodies has been further studied and new data suggest that immune response is influenced by the age at vaccination. Here, we aim to investigate the influence of both the duration and age-specificity of vaccine-induced protection on our model predictions and explore how the optimal vaccination strategy may change in the long-term. We adapted our previous model and considered plausible alternative immunization strategies, including the addition of a booster dose to the current schedule, as well as the routine vaccination of school-aged children for a range of different assumptions regarding the duration of protection. To allow for a comparison between the different strategies, we use several metrics, including the median age of infection, the number of people needed to vaccinate (NNV) to prevent one case, the age distribution of cases for each strategy, as well as the time it takes for the number of cases to start increasing after the honeymoon period (resurgence). None of the strategies explored in this work is superior in all respects. This is especially true when vaccine-induced protection is the same regardless of the age at vaccination. Uncertainty in the duration of protection is important. For duration of protection lasting for an average of 18 years or longer, the model predicts elimination of NmA cases. Assuming that vaccine protection is more durable for individuals vaccinated after the age of 5 years, routine immunization of older children would be more efficient in reducing disease incidence and would also result in a fewer number of doses necessary to prevent one case. Assuming that elimination does not occur, adding a booster dose is likely to prevent most cases but the caveat will be a more costly intervention. These results can be used to understand important sources of uncertainty around MenAfriVac and support decisions by policymakers.

## 1. Introduction

Countries in the meningitis belt of sub-Saharan Africa have been repeatedly devastated by meningitis epidemics since the early 1900s. Primarily, these epidemics are caused by the bacterium *Neisseria meningitidis* and a number of circulating meningococcal serogroups are responsible for causing disease in the meningitis belt [1]. Until 2010, the predominant serogroup responsible for frequent epidemic cycles was *N. meningitidis* serogroup A (NmA) [2]. Since the introduction of a tailor made vaccine, MenAfriVac in 2010, over 300 million 1–29 year olds have been vaccinated against NmA, resulting in a more than 99% decline in the number of confirmed group A cases in fully vaccinated populations [3]. 

We previously developed a transmission dynamic model of NmA with the aim of forecasting the relative benefits of different immunisation strategies [4]. The model highlighted the importance of a long-term vaccination strategy following the introductory mass campaigns of 1–29 year olds. Of the long-term strategies we investigated, a combination strategy of routine immunisation within the Expanded Programme on Immunisation (EPI) together with a mini catch-up, targeting children born after the introductory campaign, was the most effective. After reviewing the model findings and additional comprehensive information from clinical trials, the World Health Organisation’s recommendation for the countries of the African meningitis belt is to introduce MenAfriVac into routine immunisation programmes within 5 years after completion of the mass campaigns. The vaccine regimen is a 1-dose schedule given at 9–18 months of age. At the time of introduction into EPI, it is recommended that countries should also include a one-time catch-up campaign to immunise those born since the introductory campaigns [5]. 

One of the key assumptions in our previous work was that the duration of vaccine induced protection is the same for all ages. Due to limited data at the time, we assumed that MenAfriVac offered protection for an average of 10 years. Since then, several studies have investigated the persistence of vaccine-induced antibodies and the influence of age at vaccination. These studies provide empirical evidence on the duration of the immune response to MenAfriVac, which may be used as a proxy to the duration of protection. Correlates of protection for meningococcal disease are based upon serum bactericidal activity (SBA) [6]. The studies by White et al. [7] and Yaro et al. [8] suggest that vaccine protection is age-dependent and lasts longer for individuals targeted after the age of 2 years or 5 years, respectively. These new studies were consistent in suggesting that the duration may be age-specific, but inconsistent in their estimates of the duration.

The aim of this paper is to investigate the influence of both the duration and age-specificity of vaccine-induced protection on our model predictions and explore how the optimal vaccination strategy may change in the long-term. More specifically, we consider four scenarios that could be plausible alternative strategies to the current.

## 2. Materials and Methods

### 2.1. Model Structure

Details of the model structure have been previously published [4]. In brief, it is a compartmental model that divides the population into: susceptible state, carrier of NmA, disease due to NmA, and recovered and immune, with each of these states replicated for vaccinated and unvaccinated. 

In a modification for this paper, instead of having broad age groups, we now divided the population into annual age cohorts. A model modification was also necessary in order to simulate an age-dependent duration of protection. We added four new compartments to the previous model. These four states represent the susceptible, carriers, diseased, and recovered/immune who receive vaccination before the age of five years. A table with all of the compartments and their descriptions can be found in Appendix A, together with a flow diagram of the model.

### 2.2. Model Parameters

Demographic data for Chad were used to estimate parameters for the model. Epidemics of NmA in Chad in the pre-vaccine era occurred every 8–12 years, which is representative of the epidemiology of NmA in the African meningitis belt [9]. The introduction of MenAfriVac in that country was completed in two phases during 2011 and 2012 [10]. In the model, because there is no geographic sub-division, we assume that 50% of the target population were vaccinated in 2011 while the remaining 50% received the vaccine in 2012. 

We assumed that the coverage in Chad for routine immunisation of infants starts at 75% in 2017 and continues with annual increments of 1% until it reaches 90%. It is then assumed to stay constant for the remaining years. There is no vaccine currently being administered at 5 or 10 years of age. Hence, we explored the impact of vaccinations, assuming a coverage of 80% at these ages. Other parameters were based on the available literature wherever possible, as previously described [4]. 

To account for the uncertainty around the duration of protection, we ran each scenario outlined below under the following four different assumptions: (1) an average of 5 years duration of protection for all ages; (2) 10 years duration of protection for all ages; (3) 20 years duration of protection for all ages; and (4) 5 years duration of protection for <5 year olds and 10 years duration of protection for children at 5 years of age or older. 

### 2.3. Vaccination Strategies

We considered a range of vaccination strategies (Table 1) that were elucidated through informal discussions with colleagues at WHO, PATH, and CDC to be of interest. 

### 2.4. Model Implementation

The model is run for the time period 2010–2060 using a daily time step. For each model run, the number of cases by age is calculated per year. The average number of cases and the percentage of cases prevented by each of the strategies is calculated over 200 simulation runs per strategy. To account for the uncertainty due to the stochastic nature of the model, 95% confidence intervals were calculated using a student-t distribution as implemented by the *t*-test function in R version 3.4.2 [11].

To allow for a comparison between the different strategies, we report the time to resurgence, median age of infection, and the age distribution of cases for each strategy. As time to resurgence, we define the year in which the number of cases exceeds the threshold of 1 case per 100,000 population following the preventive campaigns. The comparison of each metric is based on non-overlapping confidence intervals. Due to the large range from 10 years to 20 years duration of protection for all ages, we also investigate the effect of all the intermediate years on the time to resurgence in a sensitivity analysis. As an additional measure of efficiency, we calculated the number of people needed to vaccinate (NNV) to prevent one case [12]. We define NNV as the total number of doses administered divided by the total number of cases prevented under each vaccination strategy over the time period under consideration. The total number of doses given for each scenario was calculated by multiplying the total number of people targeted with the assumed age-specific vaccine uptake.

## 3. Results

### 3.1. Baseline Scenario (10 Years Duration of Protection)

The model results suggest that if the assumed duration of protection is 10 years for all ages, routine immunization aimed at schoolchildren is not better than routine immunization of 1-year-old children. However, switching the age at vaccination from 12 months to 5 years is the single dose strategy with the lowest average number of cases predicted, albeit there is a 5-year period with two doses. The numerical results for the different strategies are given in Appendix B. The strategy that leads to the largest number of cases averted is the Booster strategy with a 79.3% (CI: 78.7–79.8%) predicted overall reduction, relative to a 66.8% (66.2–67.4%) predicted reduction if the current strategy remains unchanged until 2060, but the NNV is much higher (Table A2).

### 3.2. Time to Resurgence

The model predicts that when the assumed duration of protection is 12 years or shorter for all ages, a resurgence always follows the initial mass campaigns (Figure 1). The size of the peak as well as the year of resurgence both depend on the schedule and the duration of protection. The longer the duration of protection, the longer the honeymoon period is. No resurgence was seen in model runs (i.e., 100% of the 200 simulations result in elimination) when vaccine-induced protection was assumed to last for an average of 18 years or longer. For an assumed duration of protection of 16 years for all ages, 69.5% of the simulation runs resulted in elimination after the mass campaigns. Summary statistics showing the year the disease incidence exceeds the threshold of one case per 100,000 population for duration of protection between 10 and 20 years can be seen in Table A3 in Appendix B.

If we assume that duration of protection is 5 years regardless of the age at vaccination, the model predicts that the number of cases will start increasing only 10 years after the introduction of MenAfriVac, compared to ~17 years of honeymoon period when duration of protection is 10 years. Earlier resurgence does not necessarily translate to a larger number of total cases (Figure 2). 

### 3.3. Burden of Disease 

Of the strategies considered, the Booster strategy resulted in the fewest cases across all different assumptions regarding the duration of protection (Figure 3, Table 2). Taking into consideration only the single-dose schedules, the model results suggest that if the duration of protection is assumed to be the same for everyone regardless of at what age they are targeted, routine immunization at 12 months of age (EPI@12m) is similar to routine immunization at older ages (EPI@5y and EPI@10y). There is considerable overlap in the results but strategy Switch is the strategy with the lowest average number of total cases predicted (Figure 3). However, assuming that vaccination of 1-year-old leads to a shorter duration of antibody persistence compared to vaccination at older ages, strategies EPI@5y and EPI@10y result in a lower number of predicted cases compared to the EPI@12m strategy. 

Vaccination programmes raise the average age of infection since vaccinated children are protected against disease. Routine immunization at 10 years (EPI@10y) is associated with the lowest median age of infection as it results in a large number of unprotected children at a very young age leading to a large number of cases in the under 10-year-olds (Figure 4).

The strategy with the highest median age of infection is the routine immunization targeting children on their first birthday, with or without a booster dose when they turn 5 years of age. Anyone can develop invasive meningococcal disease, but rates of disease are higher in children under the age of 5 years [13]. However, carriage prevalence is higher in individuals aged 5–19 years [14]. Routine immunization of 1-year-old children leads to waning of vaccine protection by the teenage years, when there is still heightened risk of meningitis. A booster dose extends the protection until the individuals age into a lower risk age group, which in turn results in a decreased transmission. 

## 4. Discussion

At the time of developing our previous model, data on the duration of vaccine-induced protection was limited. We based our assumption of an average of 10 years duration of protection on findings from unpublished trials and expert opinion. The initial mass campaigns in the countries of the meningitis belt started taking place in 2010, but vaccination in children under the age of 12 months did not start before 2016. Here, we update our previous model to take into account findings from two recent studies, suggesting that protection lasts longer in individuals receiving MenAfriVac after the age of two years [6,7]. We used this updated model to assess the impact of a set of new vaccination strategies and compared them to the current strategy followed by African countries, since 2015.

Assuming that the duration of protection is at most 10 years, model results suggest that meningococcal disease cannot be eliminated within the first 50 years after the initial vaccination by the current or new strategies explored. On the contrary, provided that high antibody levels persist for an average of 20 years, all strategies, including the current, result in a possible elimination of NmA cases since there are no predicted cases until at least 2060.

As a long-term strategy, in the absence of any catch-up campaigns, routine vaccination of 10 year olds would lead to the smallest average number of cases. However, including the campaigns, in the case of determining which strategy leads to the least number of cases, assuming that the duration of protection is the same across all ages, no single-dose strategy is superior to the rest as there was considerable overlap in the results. This is due to the mini catch-up campaign, which is part of only the current strategy (EPI@12m) and not the other two (EPI@5y and EPI@10y). The main difference in the results comparing the strategies is in the age distribution of cases. Reductions in the number of cases in one age group results in a rise of cases in another age group. Routine vaccination at 12 months offers better protection in young children, whereas vaccination at older ages reduces disease burden in adolescents and young adults. The risk of developing at least one major sequelae after meningococcal meningitis is higher in children under the age of 5 years [15]. In this study, we do not calculate Disability Adjusted Life Years (DALYs), where an age-specific weight may be appropriate. Assuming that vaccine protection is short-lived in children under the age of 5 years, the model suggests that it would be wiser to change the target age of routine immunization from 12 months to 5 years provided that coverage is at least 50%. 

Routine immunization at 10 years of age (EPI@10y) is consistently the most effective strategy across all different assumptions about the duration of vaccine protection in terms of the number of people needed to vaccinate (NNV). This is due to the small number of doses administered, calculated based on Chad’s population demography. The high annual growth rate of the country results in a triangle-shaped age distribution with the number of individuals declining with age. The strategy associated with the highest NNV is the strategy with the additional booster dose since the number of doses is almost double that of the rest of the strategies. NNV is widely used in the scientific literature. The nature of the disease (endemic, epidemic, high/low $R_{0}$) as well as the way NNV is calculated can produce biased results [12] and, thus, caution should be taken when interpreting results or comparing NNVs with other diseases in the scientific literature. However, the highest NNV value of 485 produced by the simulations for the Booster strategy is far superior to NNV 2800–3700 estimated by Trotter et al. [16] when evaluating the response thresholds for reactive vaccination campaigns.

This is the first model to explore the potential benefits of targeting schoolchildren for routine immunization with MenAfriVac. As in all mathematical models, there is uncertainty around the model structure and certain key model parameters. The results from this work were generated using demographic data from Chad, a country lying entirely in the meningitis belt and which suffered from epidemics every 8 to 12 years before the introduction of MenAfriVac in 2011 [9]. The same structure is used to model different countries across the belt; here, we chose Chad as a typical example, but given that country-specific demography is not substantially different, we believe the results are more broadly generalisable to other meningitis belt countries. A number of key parameters, such as the transmission rate and the duration of natural immunity remain unknown; therefore, were kept the same as in the original study, allowing for a more direct comparison. Mixing parameters are also important in age-structured models. The carriage prevalence produced by our model is consistent with contact studies in Africa, in which the highest intensity of contacts is observed in 5–15 year olds [17]. 

We used several metrics to compare the different strategies qualitatively and quantitatively. None of the strategies explored in this work is superior in all respects. This is especially true when vaccine induced protection is the same regardless of the age at vaccination. Immunising infants (EPI@12m) offers protection to young children and raise the median age of infection. However, the NNV to prevent one case is higher than the NNV to prevent one case when EPI targets 10-year-olds (EPI@10y). Leaving children up to the age of 10 years unprotected, however, results in more cases in younger ages and less in older age groups. The Booster strategy may result in the least number of cases but it is the most costly intervention since it needs two doses and therefore we assume approximately double the cost of the others. A possible change in the current immunization schedule would have to be based in the prioritization of all the above factors. 

The uncertainty around the assumptions regarding the duration of protection was also explored in another mathematical model forecasting the impact of MenAfriVac vaccination by Jackson et al. [18]. In their study, they mainly focused on updating and validating their previous model in light of newly data [19]. In contrast to our work, Jackson et al. assumed that routine vaccination solely targets 9 months old children. They also explored the benefits of adding a booster dose at 10 years of age in a sensitivity analysis. Despite their structural differences, both models highlight the critical need for a long-term immunization strategy to sustain low levels of infection as well as the importance of continuous updating of models when new data become available.

Since the start of immunization with MenAfriVac, there has been an increased disease incidence caused by serogroups other than serogroup A. A new pentavalent vaccine is being currently developed with the expectation of licensure by end of 2022 [20]. In order to estimate the impact of introducing this new pentavalent vaccine in an already vaccinated population, a more robust study, including a multi-serogroup model, should be performed. This will involve a number of new unknown parameters and further increase the complexity of the model structure. Yaesoubi et al. [21] developed a transmission dynamic model to investigate the cost-effectiveness of alternative vaccination strategies using the novel multivalent vaccine. They concluded that the inclusion of a catch-up campaign with the novel vaccine would be a cost-effective way to further reduce the meningococcal disease burden. 

Despite the limitations of this work, and the uncertainty surrounding the introduction of the pentavalent vaccine in the countries of the African meningitis belt, this analysis and the conclusions drawn can be used in the future by policymakers to understand the importance of the duration of vaccine protection and support decision making around vaccine scheduling, such as a shift to routine immunization at an older age or the addition of a booster dose. This change can either be the addition of a booster dose at a later age or simply the age of the primary dose. The aim of this study is to identify the optimal way to maintain the success of MenAfriVac in reducing the number of MenA cases in the long-term. Additional work on the feasibility and cost-effectiveness of policy changes is also essential. In the future, with the advent and rollout of affordable multivalent vaccines, protection against NmA and other serogroups will be enhanced. 

## 5. Conclusions

Models can be useful in investigating a range of assumptions and a variety of vaccine strategies. Further empirical studies of the duration of protection (or the duration of the immune response) following MenAfriVac will help to decrease uncertainty about the optimal vaccination policy. 

## Figures and Tables

**Figure 1 microorganisms-09-00461-f001:**
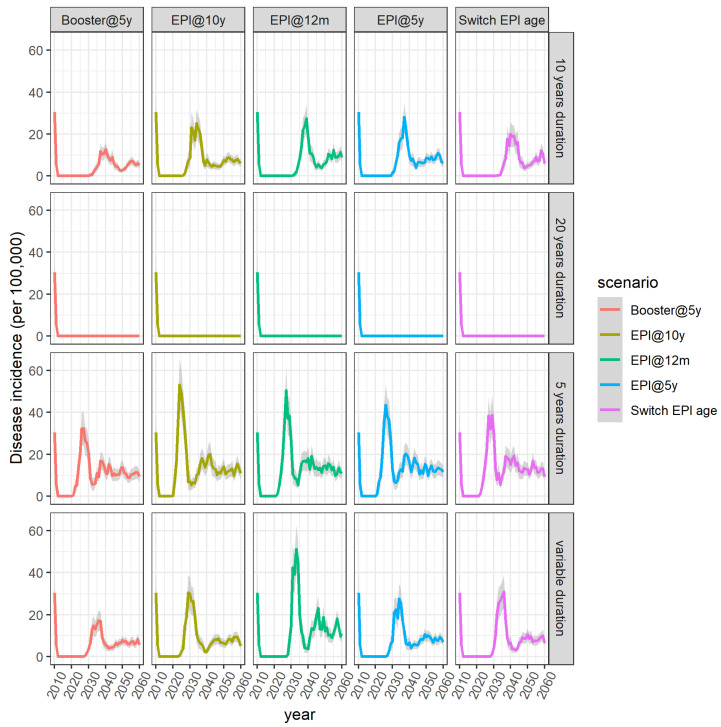
Average disease incidence across the different vaccination scenarios and across the different assumptions regarding the duration of vaccine-induced protection. Shaded areas represent the 95% confidence intervals.

**Figure 2 microorganisms-09-00461-f002:**
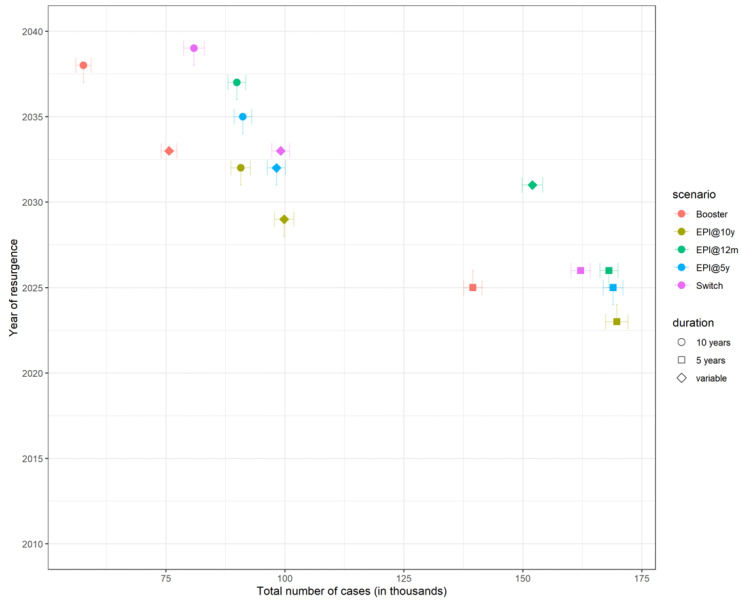
Total number of cases plotted against the year of resurgence across all scenarios and all assumptions regarding duration of protection and coverage. Each strategy is represented with a different colour and each assumption about the duration of protection is represented with a different symbol shape. Note that 20 years duration of protection is not shown. Error bars show the 95% confidence interval.

**Figure 3 microorganisms-09-00461-f003:**
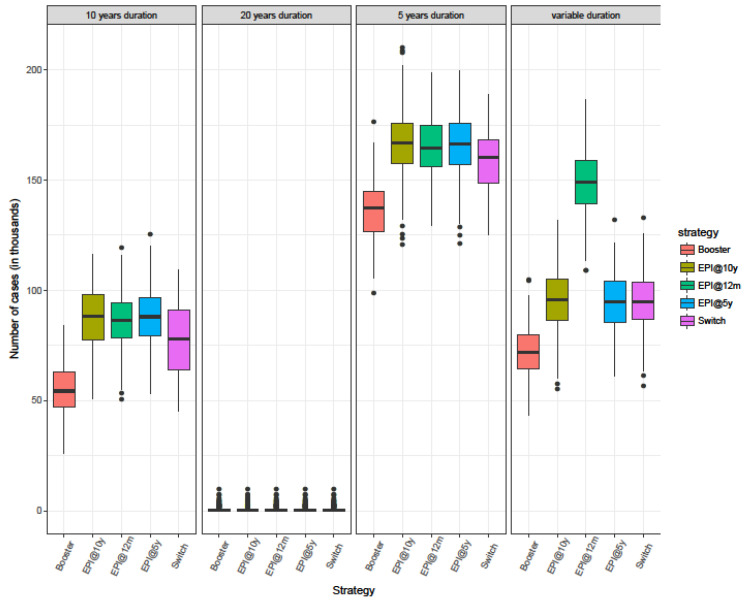
Box plot showing the median, interquartile range, and full range of the predicted total number of cases for different immunisation strategies in the time period 2010–2060 from 200 simulation runs.

**Figure 4 microorganisms-09-00461-f004:**
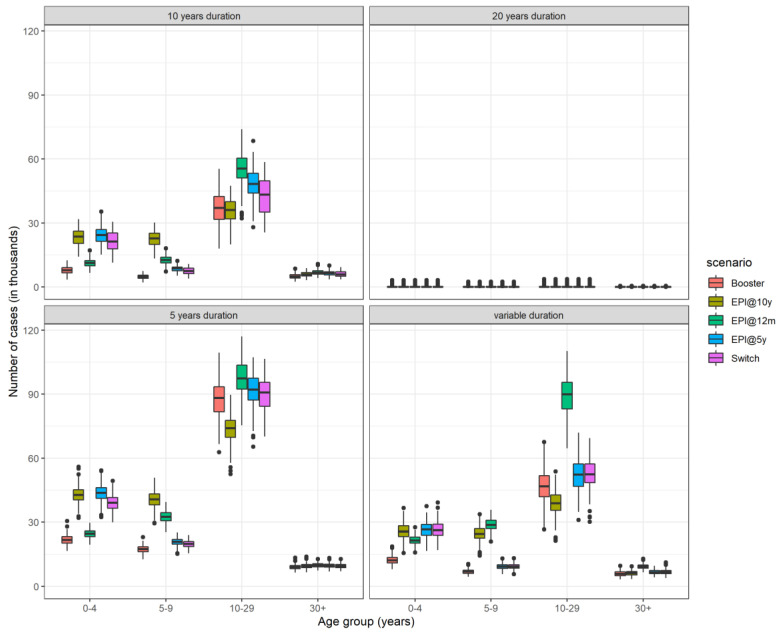
Box plot showing the median, interquartile range, and full range of the total number of cases by age group from 200 simulation runs aggregated over the time period 2010–2060.

**Table 1 microorganisms-09-00461-t001:** Vaccination strategies considered in the model.

Strategy	Introduction	Catch-Up Campaign	Routine Immunisation	Assumed Coverage for EPI
EPI@12m (Current strategy)	2011–2012: 1–29 years old	2017: 1–6 years old	2017–2060: at 12 months	75% in 2017 and annual increments of 1% until it reaches 90%
EPI@5y	2011–2012: 1–29 years old	None	2017–2060: at 5 years	80%
EPI@10y	2011–2012: 1–29 years old	None	2022–2060: at 10 years	80%
Booster	2011–2012: 1–29 years old	None	2017–2060: at 12 months and 5 years	75% in 2017 and annual increments of 1% until it reaches 90% for 12 month olds 80% for 5 year olds
Switch	2011–2012: 1–29 years old	2017: 1–6 years old	2017–2021: at 12 months 2022–2026: at 12 months and 5 years 2027–2060: at 5 years	75% in 2017 and annual increments of 1% until it reaches 90% for 12 month olds 80% for 5 year olds

**Table 2 microorganisms-09-00461-t002:** Number of doses given, number of cases predicted and averted, and number of doses needed to prevent one case for each immunization strategy for the time period 2011–2060. All of the numbers, apart from the number of people needed to vaccinate (NNV), are in the thousands. Averages across 200 simulation runs.

Strategy	Duration of Protection	# of Doses	Cases	Cases Prevented	NNV
**EPI@12m**	10 years	42,034	86.36	174.39	241
**EPI@5y**	10 years	33,878	87.62	173.13	196
**EPI@10y**	10 years	30,024	87.17	173.58	173
**Switch**	10 years	39,961	77.34	183.41	218
**Booster**	10 years	63,882	54.09	206.66	309
**EPI@12m**	20 years	42,034	0.65	260	162
**EPI@5y**	20 years	33,878	0.65	260	130
**EPI@10y**	20 years	30,024	0.71	260	115
**Switch**	20 years	39,961	0.65	260	154
**Booster**	20 years	63,882	0.65	260	246
**EPI@12m**	5 years	42,034	164.53	96.22	437
**EPI@5y**	5 years	33,878	165.41	95.34	355
**EPI@10y**	5 years	30,024	166.19	94.56	317
**Switch**	5 years	39,961	158.59	102.16	391
**Booster**	5 years	63,882	135.92	124.83	512
**EPI@12m**	Age-specific	42,034	148.6	112.3	374
**EPI@5y**	Age-specific	33,878	94.66	166.1	204
**EPI@10y**	Age-specific	30,024	96.28	164.47	183
**Switch**	Age-specific	39,961	95.56	165.19	242
**Booster**	Age-specific	63,882	72.09	188.66	339

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
