# Peer review of "Understanding the Role of Duration of Vaccine Protection with MenAfriVac: Simulating Alternative Vaccination Strategies"

_microorganisms, 2021, doi:10.3390/microorganisms9020461_

Round 1

Reviewer 1 Report

This paper considers the impact of different MenA conjugate vaccine immunization strategies in sub-Saharan Africa by mathematical modelling.

Major comments

  1. Various conclusions have been drawn comparing the different immunization strategies – however I could not see that the strategies were formally compared statistically to identify if these were significant differences for any of the outcomes. Was this based on non-overlapping CIs or some other approach?
  2. Page 9 – given the differences in likely age group when disease will occur in the different models, comment should be made on the impact on the healthcare systems considering the different severity of disease in different age groups. For example, younger children at higher risk of long-term neurological complications.
  3. Discussion - How are the results generalizable across the meningitis belt? Can the epidemiology in Chad be compared to other countries to at least make some comment on this?

Minor comments

  1. The abstract would be clearer if there was brief introduction of the current vaccine schedule for MenAfriVac, before going into alternative strategies. In addition the specific results are not clear from the abstract.
  2. Page 1 – should state that MenAfriVac is a MenA-TT conjugate vaccine and the dosing regimen used in the mass campaigns and planned subsequently.
  3. Page 2 – should also be mention as to how the immunological correlate of protection based on SBA has been used to infer duration of protection.
  4. Page 4 and Abstract: define ‘resurgence’
  5. Page 9 – cost is briefly mentioned – could the authors quantify this?

Author Response

Please note that original comments are in black and our responses noted in red text.

Major comments

Various conclusions have been drawn comparing the different immunization strategies – however I could not see that the strategies were formally compared statistically to identify if these were significant differences for any of the outcomes. Was this based on non-overlapping CIs or some other approach?

We used several different metrics to compare the strategies and the comparison was based on non-overlapping Cis. We added the following text in the manuscript for clarification.

Line 115-116 “The comparison of each metric is based on non-overlapping confidence intervals”

Line 246  “We used several metrics to compare the different strategies qualitatively and quantitatively”

Page 9 – given the differences in likely age group when disease will occur in the different models, comment should be made on the impact on the healthcare systems considering the different severity of disease in different age groups. For example, younger children at higher risk of long-term neurological complications.

We added a new comment on findings from a systematic review suggesting that the risk of developing a major sequelae is higher in children under the age of 5 years. DALYs are not calculated in this study but it may be advisable to use an age-specific weight for DALYs in the future which may allow for additional comparisons of vaccine strategies.  

Line 215-218 “The risk of developing at least one major sequelae after meningococcal meningitis is higher in children under the age of 5 years [15]. In this study, we do not calculate Disability Adjusted Life Years (DALYs), where an age-specific weight may be appropriate.”

Discussion - How are the results generalizable across the meningitis belt? Can the epidemiology in Chad be compared to other countries to at least make some comment on this?

We added a comment on this in the discussion.

Line 239-242 “The same structure is used to model different countries across the belt; here, we chose Chad as a typical example, but given that country-specific demography is not substantially different we believe the  results are more broadly generalisable to other meningitis belt countries”

Minor comments

The abstract would be clearer if there was brief introduction of the current vaccine schedule for MenAfriVac, before going into alternative strategies. In addition the specific results are not clear from the abstract.

We added the following text.

Line 11-13 “Our findings suggested that the most effective strategy in maintaining disease control was the introduction of MenAfriVac into the Expanded Programme on Immunisation (EPI). This strategy is currently being followed by the countries of the meningitis belt.”.

Page 1 – should state that MenAfriVac is a MenA-TT conjugate vaccine and the dosing regimen used in the mass campaigns and planned subsequently.

Line 54-56 “The vaccine regimen is a 1-dose schedule given at 9-18 months of age. At the time of introduction into EPI, it is recommended that countries should also include a one-time catch-up campaign to immunise those born since the introductory campaigns”

Page 2 – should also be mention as to how the immunological correlate of protection based on SBA has been used to infer duration of protection.

Line 63 “.Correlates of protection for meningococcal disease are based upon serum bactericidal activity (SBA) [6]”

Page 4 and Abstract: define ‘resurgence’

Line 19-22 We rephrased. New text now reads: “To allow for a comparison between the different strategies, we use several metrics, including the median age of infection, the number of people needed to vaccinate to prevent one case (NNV), the age distribution of cases for each strategy as well as the time it takes for the number of cases to start increasing after the honeymoon period (resurgence).”

Line 114-115 Added an extra sentence clarifying. “As time to resurgence, we define the year in which the number of cases exceeds the threshold of 1 case per 100,000 population following the preventive campaigns.”

Page 9 – cost is briefly mentioned – could the authors quantify this?

The booster strategy requires 2 doses so we assume that this strategy will be approximately double the cost of others. Line 254-256 “The Booster strategy may result in the least number of cases but it is the most costly intervention since it needs 2 doses and therefore we assume approximately double the cost of the others.”

We thank you for all the helpful comments. We would be glad to respond to any further questions and comments that you may have.

Reviewer 2 Report

I have gone through the manuscript titled " Understanding the role of duration of vaccine protection with MenAfriVac: Simulating alternative vaccination strategies" It is well performed and presented as a study. However, before acceptance of the current version authors need to revise substantially as suggested below: 

Abstract

Other important analyses carried out in this manuscript should be included in the abstract.

Introduction

The introduction is not covered the story well. The introduction should cover the literature on this topic as well. Also, recent references should be included.

Materials and methods

Need more details for experiment

Results

The results are clear and important.

Discussion

The discussion is somewhat well-written but still needs improvement.

The current data should be compared with previously published findings and how these new findings support the research question.

Conclusions need improve.

Line 53-65, please rephrase it.

Line 195-206, please rephrase it.

Line 256-264, please rephrase it.

English writing should be checked by a native English speaking expert.

Author Response

We thank you for your comments. We have made certain changes in the introduction and discussion sections following the advice from the other reviewers as well to make our points clearer.  

Reviewer 3 Report

It would be interesting to include in the paper some data on the assessment of seroconversion following vaccination with MenAfriVac.  Taking into account that the vaccine was introduced into use already in 2010, it can be expected that the immunological status of people who have been immunized with it is known. By recalling these data, reader could better understand why the authors of the presented work use the statement: "(...) assumptions: (1) an average of 5 years duration of protection for all ages, (2) 10 years duration of protection for all ages, (...)" especially since the authors write that "Due to limited data at the time, we assumed that MenAfriVac offered protection for an average of 10 years". Perhaps it should be explained why 11 years after the vaccine introduction, the data in this scope are limited. Of course, the text contains appropriate and numerous references, but for a more complete message it would be worth considering supplementing the manuscript with the indicated information.
Since the authors write that in the vaccinated population as much as 99% decrease in N. meningitidis serogroup A cases is observed, the aim of the study should be more widely/prospectively expressed, is this about total elimination of NmA infections by changing the vaccination schedule?
Notwithstanding the above, I recommend the paper to be published in the journal after minor improvements as it was suggested. 

Author Response

Please note that original comments are in black and our responses noted in red text.

It would be interesting to include in the paper some data on the assessment of seroconversion following vaccination with MenAfriVac.  Taking into account that the vaccine was introduced into use already in 2010, it can be expected that the immunological status of people who have been immunized with it is known. By recalling these data, reader could better understand why the authors of the presented work use the statement: "(...) assumptions: (1) an average of 5 years duration of protection for all ages, (2) 10 years duration of protection for all ages, (...)" especially since the authors write that "Due to limited data at the time, we assumed that MenAfriVac offered protection for an average of 10 years". Perhaps it should be explained why 11 years after the vaccine introduction, the data in this scope are limited. Of course, the text contains appropriate and numerous references, but for a more complete message it would be worth considering supplementing the manuscript with the indicated information.

Vaccination of children under the age of 12 months only started in 2016 and the data are still limited. The two studies that are referenced in the manuscript provide different estimates on the duration of protection and this is the basis for our assumptions in the current paper. We added the following text in the discussion.

Line 194-198 “At the time of developing our previous model, data on the duration of vaccine-induced protection was limited. We based our assumption of an average of 10 years duration of protection on findings from unpublished trials and expert opinion. The initial mass campaigns in the countries of the meningitis belt started taking place in 2010 but vaccination in children under the age of 12 months did not start before 2016.”

Since the authors write that in the vaccinated population as much as 99% decrease in N. meningitidis serogroup A cases is observed, the aim of the study should be more widely/prospectively expressed, is this about total elimination of NmA infections by changing the vaccination schedule?

Line 279-280 “The aim of this study is to identify the optimal way to maintain the success of MenAfriVac in reducing the number of MenA cases in the long-term.”

Reviewer 4 Report

one thing of editing, to keep the intend in all document and space between lines.

Author Response

We thank you for your comment.